# PSMA-PET/MRI-Based Focal Dose Escalation in Patients with Primary Prostate Cancer Treated with Stereotactic Body Radiation Therapy (HypoFocal-SBRT): Study Protocol of a Randomized, Multicentric Phase III Trial

**DOI:** 10.3390/cancers13225795

**Published:** 2021-11-18

**Authors:** Constantinos Zamboglou, Simon K. B. Spohn, Sonja Adebahr, Maria Huber, Simon Kirste, Tanja Sprave, Christian Gratzke, Ronald C. Chen, Ernst Günther Carl, Wolfgang A. Weber, Michael Mix, Matthias Benndorf, Thomas Wiegel, Dimos Baltas, Carolin Jenkner, Anca L. Grosu

**Affiliations:** 1Department of Radiation Oncology, University Medical Center Freiburg, Faculty of Medicine, University of Freiburg, 79106 Freiburg, Germany; constantinos.zamboglou@uniklinik-freiburg.de (C.Z.); sonja.adebahr@uniklinik-freiburg.de (S.A.); simon.kirste@uniklinik-freiburg.de (S.K.); Tanja.Sprave@uniklinik-freiburg.de (T.S.); anca.grosu@uniklinik-freiburg.de (A.L.G.); 2German Cancer Consortium (DKTK), Partner Site Freiburg, 79106 Freiburg, Germany; 3Berta-Ottenstein-Programme, Faculty of Medicine, University of Freiburg, 79110 Freiburg, Germany; 4German Oncology Center, European University of Cyprus, Limassol 4108, Cyprus; 5Clinical Trials Unit, Faculty of Medicine, Medical Center, University of Freiburg, 79110 Freiburg, Germany; Maria.huber@uniklinik-freiburg.de (M.H.); carolin.jenkner@uniklinik-freiburg.de (C.J.); 6Department of Urology, Faculty of Medicine, Medical Center, University of Freiburg, 79106 Freiburg, Germany; christian.gratzke@uniklinik-freiburg.de; 7Department of Radiation Oncology, University of Kansas Cancer Center, Kansas City, KS 66160, USA; rchen2@kumc.edu; 8Bundesverband Prostatakrebs Selbsthilfe e.V., 53111 Bonn, Germany; guenther.carl@prostatakrebs-bps.de; 9Department of Nuclear Medicine, Klinikum Rechts der Isar, School of Medicine, Technical University of Munich, 81675 Munich, Germany; w.weber@tum.de; 10Department of Nuclear Medicine, Faculty of Medicine, Medical Center, University of Freiburg, 79106 Freiburg, Germany; michael.mix@uniklinik-freiburg.de; 11Department of Radiology, Faculty of Medicine, Medical Center, University of Freiburg, 79106 Freiburg, Germany; matthias.benndorf@uniklinik-freiburg.de; 12Department of Radiation Oncology, University Hospital Ulm, 89081 Ulm, Germany; thomas.wiegel@uniklinik-ulm.de; 13Division of Medical Physics, Department of Radiation Oncology, Medical Center, Faculty of Medicine, University of Freiburg, 79106 Freiburg, Germany; dimos.baltas@uniklinik-freiburg.de

**Keywords:** prostate cancer, stereotactic body radiotherapy, SBRT, ultra-hypofractionation, moderate hypofractionation, focal dose escalation, mpMRI, PSMA-PET, clinical trial

## Abstract

**Simple Summary:**

Prostate cancer (PCa) is the most diagnosed malignancy of men worldwide, and radiotherapy (RT) is a main treatment option for patients in all disease stages. Modern diagnostic imaging and RT techniques enable an effective and safe delivery of high RT doses and a reduction of treatment time. We present a study protocol of a phase III trial implementing most modern imaging techniques and RT approaches. We compare the standard RT treatment with an RT concept of reduced treatment time and focal dose escalation based on the patient’s individual tumor morphology. This HypoFocal-SBRT trial aims to individualize treatment of PCa patients.

**Abstract:**

Technical advances in radiotherapy (RT) treatment planning and delivery have substantially changed RT concepts for primary prostate cancer (PCa) by (i) enabling a reduction of treatment time, and by (ii) enabling safe delivery of high RT doses. Several studies proposed a dose–response relationship for patients with primary PCa and especially in patients with high-risk features, as dose escalation leads to improved tumor control. In parallel to the improvements in RT techniques, diagnostic imaging techniques like multiparametric magnetic resonance imaging (mpMRI) and positron-emission tomography targeting prostate-specific-membrane antigen (PSMA-PET) evolved and enable an accurate depiction of the intraprostatic tumor mass for the first time. The HypoFocal-SBRT study combines ultra-hypofractionated RT/stereotactic body RT, with focal RT dose escalation on intraprostatic tumor sides by applying state of the art diagnostic imaging and most modern RT concepts. This novel strategy will be compared with moderate hypofractionated RT (MHRT), one option for the curative primary treatment of PCa, which has been proven by several prospective trials and is recommended and carried out worldwide. We suspect an increase in relapse-free survival (RFS), and we will assess quality of life in order to detect potential changes.

## 1. Introduction

The HypoFocal-SBRT study is a randomized controlled phase III trial that will tackle one of the current standard radiotherapy (RT) approaches for primary localized prostate cancer (PCa) by integrating two modern RT concepts in the experimental arm of the study: (i) stereotactic body RT (SBRT), and (ii) focal dose-escalated RT based on advanced medical imaging. 

RT delivery concepts for primary PCa have significantly been changed by treatment planning and delivery advances with safe delivery of high RT doses and a reduction of treatment time via hypofractionation. In moderately hypofractionated RT (MHRT), RT doses of approximately 2.5–3.1 Gy per fraction are delivered. MHRT is considered as one recommended curative treatment option for primary PCa RT, due to the published results of several randomized controlled trials (RCTs) [1,2,3,4,5]. However, a further increase of dose per fraction in ultra-hypofractionated or SBRT bears potential therapeutic benefits due to the low α/β ratio of prostate cancer [6]. Two RCTs reported good safety profile with doses of 6.1–7.25 Gy per fraction [7,8]. Additionally, for primary PCa a dose–response relationship with improved tumor control rates at higher doses has been demonstrated [9,10,11]. Overcoming the issue of limited possibilities for dose escalation to the whole prostate, concepts of focal RT dose escalation have been implemented in RT practice, enabling the macroscopic intraprostatic tumor mass (ITM) to be irradiated under maintenance of dose constraints to adjacent organs at risk (OAR). Recently, the long term results of the FLAME trial demonstrated a 7% improvement in biochemical recurrence free survival (bRFS) for dose escalation based on multiparametric magnetic resonance tomography (mpMRI) [12]. MpMRI presents the gold standard for PCa detection [13] and ITM delineation, but it is limited by complex interpretation [14] and underestimation of the true tumor mass [15,16]. Positron emission tomography targeting the prostate specific membrane antigen (PSMA-PET) has been implemented in the staging of primary PCa [17], and its excellent performance for ITM detection has been demonstrated by several studies [18,19,20,21,22,23]. MpMRI and PSMA-PET give complementary information [20], and combined usage of both imaging modalities increases sensitivity and specificity for ITM delineation significantly [24,25]. Consequently, focal dose escalation considering the individual geometric distribution of ITMs based on mpMRI and PSMA-PET might further improve oncological outcomes due to enhanced coverage of the intraprostatic tumor burden [11]. Therefore, the HypoFocal-SBRT study integrates advanced biological imaging techniques for boosting the ITM (mpMRI and PSMA-PET) with high precision RT concepts in terms of SBRT in the experimental arm for patients with primary localized unfavorable intermediate- and high-risk PCa compared to a homogeneously delivered MHRT to the entire prostatic gland in the control arm of the study.

Thus, the experimental arm of the HypoFocal-SBRT trial combines (i) state of the art diagnostic imaging (mpMRI and PSMA-PET/CT) in order to accurately depict the ITM and to exclude distant metastases, with (ii) high-precision ablative RT in terms of SBRT in order to (i) improve tumor control without increasing toxicity and/or impairing quality of life (QoL), but to (ii) reduce the treatment time and thus increase patients’ convenience. 

## 2. Methods and Design

### 2.1. Study Design and Legal Aspects

The HypoFocal-SBRT trial is a multicenter randomized controlled trial with approximately 20 participating centers in Germany, Austria, Switzerland and Cyprus. Included eligible patients are randomized 1:1 (see below) into either the experimental SBRT arm (receiving 30 Gy and 35 Gy in 5 fractions to the seminal vesical and the entire prostatic gland, respectively, as well as a simultaneous integrated boost (SIB) to the ITM of 40–42 Gy depending on ITM volume and planning feasibility) or into the control MHRT (receiving MHRT 46.4 Gy and 60–62 Gy in 20 fractions for unfavorable intermediate- and high-risk PCa to the seminal vesicles and the prostate, respectively (NCCNv2.2021)). The study protocol was approved by the Medical Ethics Committees of all participating centers. Written informed consent will be obtained from all patients. The study was additionally submitted to the German Ministry for Radiation Protection (BfS). The study is registered in the German Clinical Trials Register (DRKS00022915) and “clinicaltrials.gov” registries. The study protocol was developed in cooperation with national and international expert panels as well as a representative of the PCa national self-help group. 

### 2.2. Justification of the Treatment Choice for the Experimental and the Control Arm 

The following current evidence was the base to use SBRT in the experimental arm of the study: First phase II and III studies suggested that a homogeneous SBRT to the entire prostatic gland with 35–42.7 Gy in 5–7 fractions has equivalent failure rates 5 years after treatment compared to conventional RT (CRT) for intermediate- and high-risk PCa [7,8,26]. Although the HYPO-RT-PC trial reported more pronounced gastrointestinal (GI) and genitourinary (GU) early side effects for SBRT (42.7 Gy in 7.1 Gy, EQD2 = 91 Gy, α/β = 1.6 Gy) compared to conventional fractionation with 2 Gy (CRT), late toxicity was essentially identical in both treatment groups [8]. The PACE B trial reported similar early gastrointestinal (GI) and genito-urethral (GU) toxicity rates for SBRT (36.25 Gy in 7.25 Gy, EQD2 = 89 Gy, α/β = 1.6 Gy) and CRT [7]. Thus, the NCCNv2.2021 guidelines recommend SBRT (42.7 in 7.1 Gy or 36.25–40 in 7.25–8 Gy) as a treatment option for patients with localized PCa. Considering (i) the potential biological benefit due to the proposed low alpha/beta value of primary PCa lesions [6], (ii) increased precision in RT treatment delivery enabling delivery of ablative doses and sparing of the surrounding normal tissue and (iii) the increased convenience for the patients (only 5 fractions of RT), we decided to use SBRT for focal dose escalation in the experimental arm. 

The following current evidence led to our considerations regarding focal RT within the experimental arm: For RT of primary PCa, a dose–response relationship between RT dose and tumor control rates has been postulated [9]. Moreover, local recurrences after RT often occur at the site of the primary macroscopic tumor, and RT under dosage to these areas is associated with poorer biochemical control [27]. A reduction of local and biochemical failure is associated with decreased risk for distant metastases and PCa-associated death [28,29]. However, due to the proximity to the rectum and bladder, an unlimited dose escalation to the whole prostatic gland with external beam RT would entail a high risk of severe toxicity [30]. This issue may be tackled by focal RT dose escalation approaches by irradiating the macroscopic ITM with a high dose, while the dose constraints to the rectum and bladder are maintained. In silico planning studies from our group suggested that escalating the RT dose to the ITM defined by mpMRI and PSMA PET/CT might increase biochemical control [11,31] without increasing the risk of bladder and rectum toxicities. Consequently, first retrospective and phase I/II studies have been reported on focal RT. Two phase III studies (FLAME (NCT01168479) and HEIGHT (NCT01411332)) performed RT dose escalation by applying MHRT to the ITM in the experimental arm. Recent findings from the FLAME trial suggested that focal RT dose escalation results in a significant improvement in biochemical disease-free survival without a significant increase in GU and GI toxicity when compared to the standard treatment after a median follow-up time of 72 months [12]. Considering SBRT for focal dose escalation, the phase II hypo-FLAME trial [32] delivered in median 44.7 Gy to the ITM in 5 fractions. No grade 3 acute GU or GI toxicity was observed. The hypo-FLAME trial included patients with intermediate- and high-risk PCa. Herrara et al. performed a phase 1a/1b trial and delivered up to 50 Gy in 5 fractions to the ITM in 20 patients [33]. No grade 3 toxicities were reported up to 90 days after treatment. The SPARC trial delivered 44.85–48.25 Gy in 5 fractions to the ITM [34]. An interim safety analysis was performed on the first 8 patients after a median follow-up of 56 months. Acute grade 2+ GU and GI toxicities were 37.5%, respectively, while late grade 2+ GU and GI toxicities were 12.5% and 0%, respectively. In the experimental arm of the HypoFocal-SBRT study, the dose applied to the entire prostate is 35 Gy, whereas the dose applied to the ITM is 42 Gy.

In the experimental arm of the HypoFocal-SBRT study, a dose of 35 Gy will be applied to the entire prostate [18]. Compared to other SBRT focal-dose escalation studies [32,34], the applied dose to the ITM in our study is lower (40–42 Gy), considering an improved tumor coverage due to the implementation of advanced medical imaging. All previous studies performed ITM definition based on mpMRI information. However, a recent study postulated that mpMRI detects less than two-thirds of clinically significant PCa foci [35]. Our group demonstrated in histo imaging comparison studies [36,37,38,39] and is in agreement with others [25] that sensitivity and specificity are superior in PSMA PET/CT in comparison to mpMRI in ITM detection, although both imaging modalities provide complementary information. Furthermore, an in silico planning study from our group calculated significant higher tumor control probabilities when combined PSMA PET/CT and mpMRI information was considered for ITM delineation [11]. Consequently, in the HypoFocal-SBRT study, ITM definition will be performed based on PSMA PET/CT and mpMRI. 

We decided to use MHRT for the control arm, as it is recommended as one preferred treatment option for unfavorable intermediate- and high-risk PCa patients in international guidelines (e.g., NCCNv2.2021). Three phase III studies included patients with high-risk PCa and compared MHRT with CRT. The CHHiP [3] study had a non-inferiority design, and 12% of the 3162 included patients had high-risk disease: in the MHRT arm (60 Gy in 3 Gy, EQD2 = 76.7 Gy, α/β = 1.6 Gy), 129 patients with high-risk disease were included and after 5 years of follow-up (FU) time, 84% of the patients were failure free. The two other studies had a superiority design for MHRT and included 100% (n = 83) [5] and 74% (n = 299) [40] high-risk PCa patients in the MHRT arms. The HYPRO trial applied 64.6 Gy in 3.4 Gy (EQD2 = 90 Gy, α/β = 1.6 Gy), and after 5 years of FU time the failure-free survival rate was 80.5%. The Italian trial delivered 62 Gy in 3.1 Gy (EQD2 = 81 Gy, α/β = 1.6 Gy) and reported a failure-free survival rate of 72% after 10 years of FU. In conclusion, in all three studies, no statistically significant differences in failure-free survival rates between CRT and MHRT were observed (including subgroup analyses with only high-risk patients). Considering that the standard RT dose for high-risk PCa patients is up to 80 Gy (EQD2), a dose of up to 62 Gy in 20 fractions (EQD2 = 81 Gy, α/β = 1.6 Gy) [6] is recommended in the HypoFocal-SBRT study for these patients. For the CHHiP trial, only 11.7% of the patients in the 60 Gy group had RTOG grade 2+ toxicities with a very good quality of life (QoL) [3]. Another phase III trial applied MHRT with 60 Gy (PROFIT trial), and grade 3 toxicities occurred in less than 4% of the patients [4]. Consequently, considering equivalent and well tolerated treatment outcomes compared to CRT, but shorter treatment time, MHRT will be used in the control arm according to the recommendations of current international guidelines (NCCNv2.2021 or European Association of Urology (EAU) [41]) using the dose/fractionation regimen of the CHHiP and PROFIT trial. For patient safety, we implemented the updated strict dose constraints from the CHHiP trial [42].

### 2.3. Patients—Eligibility Criteria

Patients with histologically confirmed and previously locally untreated unfavorable-intermediate and high-risk adenocarcinoma of the prostate according to NCCNv2.2021 criteria will be enrolled into this trial. All patients must be staged cN0 (no pelvic lymph nodes) and cM0 (no distant metastases) with PSMA PET/CT or PET/MR and 3 Tesla MRI fulfilling standard requirements for PCa prior to inclusion into the study. The eligibility criteria are listed in Table 1.

### 2.4. Primary and Secondary Endpoints

The primary endpoint is relapse free survival (RFS) defined as time from randomization until relapse or death. Relapse is defined as biochemical recurrence according to Phoenix criteria (two consecutive increases of PSA Nadir +2 ng/mL) [43], new PCa lesions in imaging (preferred mpMRI and/or PSMA PET/CT) or commencement of any salvage therapy. For the sample size calculation, we estimated a benefit in RFS of 10% after 5 years (please see 2.6).

Secondary endpoints are the following: time to local failure (confirmed by biopsy) after randomization, metastatic free survival after randomization (all metastases must be confirmed by imaging, preferably PSMA-PET/CT or mpMR imaging) as well as overall (OS) and prostate cancer specific survival (PCSS) after randomization. Additionally, time to biochemical failure (phoenix definition) after randomization [44] will be assessed. Patient-reported acute and late quality of life (QOL) will be assessed with the Expanded Prostate Index Composite-26 (EPIC-26) Short Form and International Prostate Symptom Score (IPSS) questionnaires; and cumulative acute (during and up to 3 months after RT) and chronic (>3 months after RT) GU and GI toxicities by using the CTCAEv5.0 criteria. We will also report on feasibility and adherence to dose constraints by measuring the ratio between number of patients with fulfilled dose constraints and prescription doses/recruited patients.

### 2.5. Randomization

Prior to randomization, a re-check of important criteria and RT planning metrics is performed. For all included patients, a SBRT-treatment plan must be created, and compliance to delineation, dose prescriptions and constraints for organs at risk will be centrally evaluated within a prospective radiation quality assurance (RTQA) procedure before randomization. Thus, biases in terms of ITM distribution and feasibility of focal dose escalation are prevented. Randomization is performed by the clinical trial unit of the Medical Center—University of Freiburg stratified by center and in blocks of variable length, disclosed to the centers, in a ratio of 1:1.

### 2.6. Recruitment and Sample Size Considerations

According to the experience of an ongoing phase II study (HypoFocal, DRKS00017570) and to obtained feasibility questionnaires, there is an average presentation of 55 patients with primary PCa present at 20 RT centers in Germany/Switzerland/Austria per year, resulting in approximately 3300 patients assessed for eligibility in three years. Considering that approximately 40% of these patients will fulfil the initial screening and a willingness for participation of 1/3, 411 patients will be included before second screening prior to randomization. Approximately 10% of patients will be excluded due to nonconformity of RT planning. Consequently, 374 patients will be eligible for randomization in 3 years.

The sample size calculation is based on the primary endpoint RFS. For the control arm, the estimated RFS is based on sub-group analysis of high-risk PCa patients within trials using MHRT, which ranged between 80.5 and 84% [1,3,5]. Considering differences in applied doses, a different definition of high-risk PCa patients in the before mentioned trials and in the HypoFocal-SBRT trial and the implementation of PSMA PET to exclude metastasized patients in the HypoFocal-SBRT trial, we estimate an RFS after 5 years of 80% for patients in the control group.

For the experimental arm, the estimated benefit in RFS is based on detailed analysis of trials evaluation dose escalation or using focal dose escalated RT. Benefits for dose escalation to the whole prostate up to 79.2 Gy ranged between 7% and 11% for intermediate-risk patients and high-risk patients [45,46,47]. Recently the FLAME trial group reported a significant benefit of 7% for mpMRI-based focal dose escalation up to 123% [12]. However, the FLAME trial included intermediate-risk patients with Gleason Score 7a and did not use PSMA-PET/CT imaging for initial staging and ITM definition. For the HypoFocal-SBRT trial, we estimate an improvement of 10% in RFS after 5 years. This further improvement is based on three main assumptions: (i) Only unfavorable intermediate-risk and high-risk patients will be included in the HypoFocal-SBRT trial, which most likely benefit the most from focal dose escalation to ITMs. (ii) Due to the implementation of PSMA-PET information additionally to mpMRI to boost definition, we will achieve a higher ITM dose coverage. (iii) In the HypoFocal-SBRT study, a higher RT dose escalation with up to 140% (EQD2 equivalent dose with α/β= 1.6 Gy of 35 Gy in 5 fraction = 83.6 Gy, EQD2 equivalent dose with α/β = 1.6 Gy of 42 Gy in 5 fraction = 116.7 Gy) will be performed. 

The RFS rate of approximately 90% in the experimental group in the same time period corresponds to a hazard ratio of 2.12 of the control group compared to the intervention group [11,31]. The effect of experimental intervention will be assessed by a test at two-sided significance level of 5% and by estimation of the hazard ratio with corresponding asymptotic two-sided 95% confidence interval. The null hypothesis is rejected if the confidence interval does not contain one. Under the above assumptions, the study is planned to detect a difference between the experimental and control intervention with a power of 80%, which requires a total number of 56 events to be observed. The required number of patients to be randomized to observe this amount of events depends on the length of follow-up. With a recruitment period of 3 years, an additional follow-up period after the end of recruitment of 4 years (87 months in total: 7 years plus 3 months of initial screening) it can safely be assumed that a sufficient number of events will have been observed by the end of the trial if a total of 354 patients (177 per group) are available for analysis (nQuery Advisor 8.3.1). Furthermore, the number of drop-outs until the assessment of the primary endpoint is assumed to be at most 10 patients per arm. Therefore, 374 patients will be randomized.

### 2.7. Imaging

Standardized imaging protocols are pivotal for a geometric individualization approach by boosting the ITM. Therefore, all imaging scans (PSMA-PET, PSMA-PET/CT, PSMA-PET/MRI and mpMRI) should be conducted by using the same protocols according to the local practice, fulfilling standard requirements for PCa imaging. Additionally, PET image reconstruction should be without the use of resolution recovery algorithms, and technical specifications for MRI acquisition should match the PRADsv2.1 recommendations [48]. The quality control of the PSMA ligands must comply with local regulatory requirements, and the radiochemical purity of the final product must be ≥97%. The commonly used PSMA ligands 68Ga-PSMA11, 18F-DCFPyL, 18F-PSMA1007 and 18F-rhPSMA7.3 are acceptable for this study. Other small molecule PSMA ligands may also be considered when they have been properly validated by prior clinical studies. It is important to mention that the diagnostic imaging must be completed prior to the registration into the study.

If patients receive androgen deprivation therapy (ADT) >1 month after initial imaging, a second planning mpMRI will be performed to guarantee optimal treatment planning considering the present prostate geometry. 

Placement of fiducials and rectal spacers is highly recommended. They should be visible on CT and mpMRI imaging to guide image fusion for RT planning and image guidance. Bladder and bowel preparation encompass a partially filled bladder and a reduced rectal diameter for all patients.

## 3. Radiotherapy

### 3.1. Contouring

The following organ at risk structures (OAR) are delineated: rectum, rectum adapt (manual delineation of the mean rectal volume based on the rectum in cone-beam CT or MR images of the first three RT fractions), bladder, urethra, right and left femoral head, penile bulb, sigmoid, small bowel and for the experimental arm PRV-rectum, PRV-Rectum_adapt and PRV-urethra (PRV = OAR + 2 mm isotropic expansion). 

For contouring of the prostatic gland, the usage of mpMRI information is mandatory. The proximal 1 and 1.5 cm of seminal vesicles (SV) must be included in unfavorable intermediate-risk and high-risk patients, respectively, according to ESTRO-ACROP guidelines [49]. When the SVs are infiltrated more distal by the ITM, these areas must also be included. 

Contouring of the ITM or gross tumor volume (GTV) within the prostate must be performed by experienced readers under consideration of all available information (imaging, localization of biopsy cores, digital rectal examination). Contouring of GTV based on PSMA-PET (GTV-PET) should be performed according to two recommendations: (i) Manual delineation by scaling PET images from SUVmin–max: 0–5 for 68Ga-PSMA-11 and SUVmin-max: 0–10 for 18F-PSMA-1007. Each uptake higher than adjacent background should be considered as PCa [21,22]. (ii) Semi-automatic delineation, where the SUVmax value within each respective lesion should be calculated and 30% (range: 20–40%) of this SUVmax values should be used as a threshold (for 68Ga-PSMA-11 and 18F-PSMA-1007) according to previous studies [18,21,22,50]. GTV-PET based on other tracers should be delineated according to each centers experience. Convolutional neuronal networks can be used for GTV-delineation [51], but the contours have always been re-checked by experienced readers. Contouring of GTV based on mpMRI (GTV-MRI) should be performed in compliance with the PIRADSv2.1 criteria [48]. In case of administration of ADT, the GTV-MRI should be delineated under consideration of the pre-ADT images. Finally, all GTVs are merged into one GTV-union volume. 

For the experimental arm (SBRT) CTV1 is defined as the prostate including any extracapsular and/or intravesical PCa. CTV_SV encompasses the proximal 1 and 1.5 cm of the SV in unfavorable intermediate-risk and high-risk patients, respectively. 

For the control arm (MHRT) clinical target volumes (CTVs) are created in analogy to the ESTRO-ACROP guidelines [49]. CTV1 is created by 3 mm expansion of the prostate, including any extracapsular PCa and exclusion of rectum and bladder. CTV_SV is created by delineation of the proximal 1.4 and 2.2 cm of the SV in unfavorable intermediate-risk and high-risk patients, respectively. CTV2 is created by the addition of the prostate including any extracapsular PCa + base of the SV (proximal 0.5–1 cm) including parts with visible tumor burden. 

Creation of the planning target volumes (PTVs) is recommended as follows, but adaption to local practice is allowed: 

For the experimental arm (SBRT) PTV1 is created by the addition of CTV1 and CTV_SV and 4 mm isotropic expansion (see Figure 1a,b). PTV 2 is created by 4 mm isotropic expansion of CTV 1 (see Figure 1c). PTV3 is created by 0–2 mm isotropic expansion of GTV-union. PTV3 must not overlap with the PRV-Rectum, PRV-Rectum_adapt, Bladder or PRV-Urethra contours. In this case, the overlapping volume must be removed from PTV3 even if parts of the GTV are excluded (Figure 2a–c).

For the definition of PTV1 and PTV2 the clinical target volume CTV1 is created by delineation of the prostate including any extracapsular and/or intravesical PCa (a, dark blue). Further, the proximal 1 and 1.5 cm of the seminal vesicles should be delineated, CTV_SV (a, orange). PTV1 is defined by the union of CTV1 and CTV_SV plus the isotropic expansion of 4 mm (b, pink). PTV2 is created by the expansion of only CTV1 with an isotropic margin of 4 mm (c, turquoise).

For the definition of PTV3, the GTV-union (Figure 3, a, green), merged by using all delineated GTVs, is extended by an isotropic margin of 0–2 mm (b). Organs at risk (OARs) are delineated (a): bladder (yellow), urethra (dark yellow), rectum (orange/brown) and respective PRVs are created by isotropic expansion of 2 mm (b). The PTV-volume, overlapping with the PRV of either rectum or urethra, must be deleted from PTV3 even if parts of the GTV are excluded (c).

For the control arm (MHRT), PTV1 is created by isotropic expansion of 6 mm for CTV1 and 8 mm for CTV_SV and merging of both volumes. PTV2 is created by isotropic expansion of CTV2 with 2–6 mm according to each center’s practice.

### 3.2. Planning Procedures

To guarantee a high quality of RT delivery, it is mandatory for each center to use the most current techniques, which will be assessed during initial quality assurance. All study centers have to use intensity modulated radiotherapy (IMRT). Radiotherapy can be performed in a LINAC system, MR-LINAC system or CyberKnife System. 

In general, dose description and documentation should be performed according to ICRU report 83 for MHRT and 91 for SBRT. Especially for SBRT, planning the usage of flattering filter free (FFF) approaches is recommended.

Prescription doses for the PTVs and constraints for OARs in the experimental arm (SBRT) are provided in Table 2 and Table 3 [7,32,52,53,54]. Prescription doses for the entire prostate must be calculated in subtraction volumes of the PTV and the Boost-PTV (PTV3). Prescription doses are the following: For PTV1 (subtraction: PTV1–PTV3): 30 Gy in 6 Gy per fraction, and for PTV2 (subtraction: PTV2–PTV3) 35 Gy in 7 Gy per fraction. The prescription dose for PTV3 ranges between 40 and 42 Gy (see Figure 3). Initial planning should be performed with a prescription dose of 42 Gy and should be reduced up to 40 Gy in 1 Gy intervals when dose constraints are not met. In cases of large boost volumes (≥10 mL and/or ≥1/3 of the prostate), the dose to PTV3 must be restrained to 40 Gy in 8 Gy per fraction.

Prescription doses for the control arm (MHRT) are recommended as follows: 46.4 Gy in 2.32 Gy per fraction for PTV1, 60 Gy in 3 Gy per fraction for PTV2 in unfavorable intermediate-risk PCa and 62 Gy (range 60–63 Gy) for high-risk PCa, respectively (NCCN v2.2021). Details are shown in Table 4. Constraints for OARs for the control arm according to the CHHiP trial protocol and a recent update [42] are mandatory and provided in Table 5.

### 3.3. Image-Guided RT (IGRT) and Adaptive Planning

IGRT plays a crucial role in successful RT delivery. For all patients it is mandatory to perform an adaptive planning step (see below) by using 3D imaging (cone-beam CT, MV-CT or MR-images) of the first three fractions of RT. Then, daily image guidance is mandatory to account for interfractional movement using 2D/2D kV-imaging (only with fiducials and even then it is mandatory to perform at least one 3D image per week), cone-beam CT or MV-CT (tomotherapy). When using fiducials, matching should be performed considering the position of the fiducials, not the bone structures. If treatment time is longer than 2 min for SBRT patients [55] it is mandatory to perform intrafractional position verification and correction of position.

Adaptive planning is performed using 3D imaging (CB-CT, MV-CT or MR images) of the first three fractions of RT. The images are transferred to the treatment planning program and fused to the planning CT. Target volume coverage and bladder and bowel filling is checked carefully. Bladder and rectum are delineated in each 3D sequence, contours are transferred to the planning CT scan, and a Rectum_adapt is created manually based on average rectum filling and position. Dose volume histograms are calculated for all volumes, and their compliance is checked centrally. If all constraints are met for all volumes, RT can be continued as planned. In case of insufficient bladder and bowel preparation, the patient must be informed about possible strategies for improvement. If more than 1 mandatory constraint is not met in more than 1 volume, a re-planning must be performed optimized on the Rectum_adapt volume. Further 3D images should be evaluated in the course of RT.

### 3.4. Androgen Deprivation Therapy 

ADT will be applied considering the NCCN v2.2021 risk group to each patient. Patients with unfavorable intermediate-risk will receive short term ADT of 4 months [56,57], and patients with high- or very-high risk will receive ADT with a duration of 18 months [58]. ADT should be started max 3 months prior to RT and must be given throughout the radiation treatment course. Administration of complete androgen deprivation is recommended for a minimum of 4 months. The time gap between ADT and conduction of initial PSMA PET/CT and mpMRI or PSMA PET/MRI scans for staging must be <1 month.

### 3.5. Quality Assurance (QA) and Safety

Central QA of the image quality (planning CT, mpMRI, PET scans and cone-beam CTs), registration between the images, contours and treatment plans will be conducted using a web-based platform for all plans by the study coordinators. Only patients with centrally approved plans will be randomized in one of the treatment arms. Prospective RT QA will be conducted in three main steps. (i) Before initiation: confirmation of strict adherence to the requirements for adaptive planning and IGRT, upload of an anonymized dataset of a primary prostate cancer patient treated in the last 6 months before initiation for QA of imaging and delineation and planning of one centrally provided benchmark case will be assessed for eligibility during this initial quality assurance. (ii) During recruitment and study treatment: the RT planning procedures (initial and adaptive plans) of all patients will be centrally reviewed and (iii) the study coordinators will organize adherence to regular meetings with the participating centers to generate a platform for exchange of experiences and pitfalls occurring during conduction of the trial.

Cumulative acute safety data (including acute toxicity and acute QoL data) will be analyzed before the overall final end of the study. This safety analysis will be conducted when all data on acute toxicity and QoL are available (6 months after randomization). Adverse events (AEs), including GU and GI events, will be documented according to CTCAEv5. AEs that occur starting from the day of signed informed consent until the end of the study must be documented in the patient’s medical record and in an electronic data capture system. 

## 4. Data Analysis

The effects of control and the experimental intervention with respect to the primary endpoint RFS will be estimated and tested by Cox regression. The regression model will include treatment and study center as independent variables, as well as categorized baseline Gleason Scores (GS 6 und 7a, GS 7b, GS 8, GS 9 und 10), categorized baseline PSA (0–10, >10–20, >20), baseline tumor volume, baseline categorized cT stage (cT2, cT3a, cT3b) and NCCN-risk group (v2.2021: unfavorable-, intermediate-, high-, very high-risk). Categories will be included as binary indicator variables. As an estimate of the effect size, the hazard ratio between the two treatment arms will be given with the corresponding asymptotic two-sided 95% confidence interval. The two-sided test of the difference between the control and the experimental intervention at a significance level of 5% will be based on the corresponding asymptotic two-sided 95% confidence interval from the Cox regression model. Relapse free survival (RFS) will be analyzed irrespective of the occurrence of intercurrent events. This is consistent with the treatment policy strategy of the estimands framework.

Descriptive analyses of the secondary endpoints will be performed in similar regression models as for primary endpoint, as appropriate for the respective type of data. Time to local failure, metastatic free survival, PCSS and OS will be measured from randomization and be analyzed using Cox regression, as described for the primary endpoint. Endpoints with competing events (time to local failure (competing: distant failure, death) and prostate cancer specific survival (competing: death of any other cause)) will be estimated using the Aalen Johanson estimator. Endpoints without competing events will be estimated using the Kaplan Meier estimator. Acute und late QoL measures (EPIC-26 and IPSS) will be analyzed descriptively by treatment arm and time point using linear regression. Change from baseline will be described.

## 5. Discussion

The HypoFocal-SBRT study was designed to investigate the individualization of RT based on patient specific tumor morphology derived from mpMRI and PSMA-PET for unfavorable-intermediate and high-risk PCa patients. For this focal RT dose escalation approach, one of the most promising treatment concepts was chosen: SBRT as it enables precise delivery of ablative doses, is well tolerated and may enhance patient comfort by reducing treatment time (only 5 fractions) [7,8,59]. It should be mentioned that in parallel to the dose escalation to the ITM (with up to 42 Gy), a decent dose de-escalation (35 Gy) to the non-tumorous prostatic tissue will be performed. 

The CHHiP [3], RTOG0415 [2], HYPRO [1], PROFIT [4] and the trial by Arcangeli et al. [5] established MHRT as one of the standard therapies for primary-localized PCa. The HYPO-RT-PC [8] and the PACE-B [7] studies have demonstrated that SBRT is a safe and effective treatment option with longer follow-up (>5 years), results pending. These results led to SBRT being recommended as a treatment option for all risk stages by the NCCN (v2.2021). To prove the assumption, that the postulated low α/β value of PCa [6] improves the therapeutic ratio of hypofractionated therapy regimes, longer follow up and more randomized data must be generated. The HypoFocal-SBRT trial will only include unfavorable intermediate- and high-risk PCa patients according to NCCN criteria to obtain evidence for hypofractionated RT regimens in these patient cohorts, overcoming the issue of non-comparability of different risk stratification systems, and will collect data for moderate and ultra-hypofractionation. 

PSMA-PET/CT has been shown to be highly sensitive and specific for staging of high risk PCa patients and provides a markedly higher diagnostic accuracy than the current clinical standard (combination of CT and bone scans) [17]. However, no phase III RCT has implemented this advanced imaging as part of RT planning. It is unknown if better staging by PSMA-PET/CT improves patient relevant outcome parameters. The HypoFocal-SBRT trial will therefore investigate a new and accurately staged patient cohort, leading clinical PCa trials into the era of PSMA-PET/CT and challenging comparability with former trials. 

A dose–response relationship for RT of primary PCa has been postulated [9], but contradicting results exist whether tumor control rates are maxing out at specific doses [6,47,60,61,62,63]. The significant increase in bRFS rates by focal dose escalation favors the hypothesis that the ceiling effect can be cracked [63]. The HypoFocal-SBRT trial will perform a boost of up to 140% and will thus provide additional information about effectiveness of focal dose escalated external beam radiotherapy (EBRT) and the biological effectiveness of hypofractionation in PCa. Furthermore, evaluation of local failure rate and localization of local recurrences will demonstrate whether the postulated improved sensitivity and tumor coverage by inclusion of PSMA-PET to ITM delineation is of clinical relevance. The correlation of dose distribution with the local recurrence pattern will provide dose–effect information. Since metastases must be confirmed preferably by PSMA-PET/CT and due to the high availability of this imaging at the participating centers, this trial will give significant information about recurrence distribution patterns. The generated data of the HypoFocal-SBRT trial will furthermore enable the comparison of mpMRI and PSMA-PET for RT treatment planning, and evaluate the IGRT procedures and the safety of focal dose escalated SBRT.

The HypoFocal-SBRT study is, to the best of our knowledge, the first randomized, controlled phase III trial comparing MHRT with focal dose escalated SBRT based on mpMRI and PSMA-PET. This study aims to push the boundaries to an individualized RT and serves as a foundation for varieties of translational projects (including genomics, genomic classifiers, radiomics and artificial intelligence for risk-prediction and patients’ active participation), enriching RT of primary PCa patients on multiple dimensions.

## Figures and Tables

**Figure 1 cancers-13-05795-f001:**
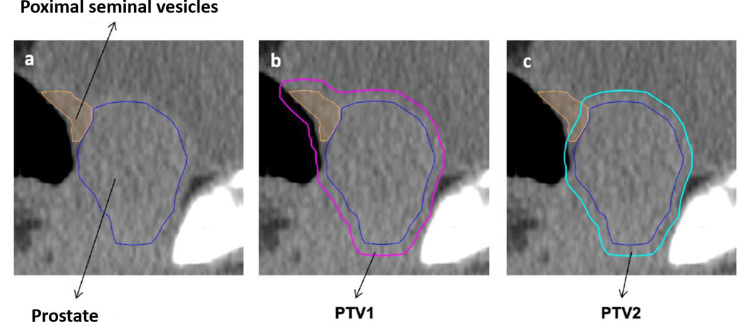
Definition of PTV1 and PTV2 for SBRT. Clinical target volume 1 (CTV1) and CTV including seminal vesicles (CTV_SV) are merged and expanded isotropic with a 4mm margin to create planning target volume 1 (PTV1) (**a**,**b**). PTV 2 is created by 4 mm isotropic expansion of CTV 1 (**c**).

**Figure 2 cancers-13-05795-f002:**
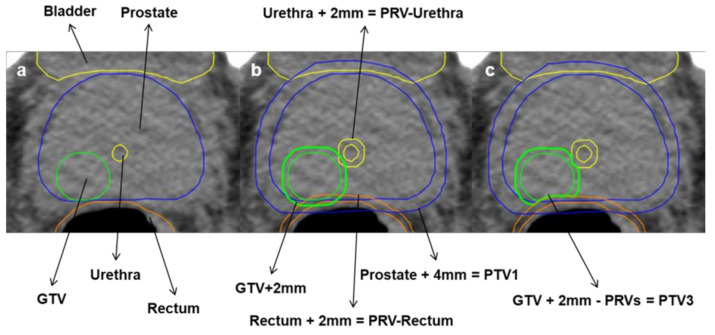
Definition of PTV3 for SBRT. Planning target volume 3 (PTV3) is created by of 0–2mm isotropic extension of the gross tumour volume (GTV) (**a**,**b**). Overlapping parts with planning organ at risk volume (PRV) of rectum and urethra are removed from PTV3 (**c**).

**Figure 3 cancers-13-05795-f003:**
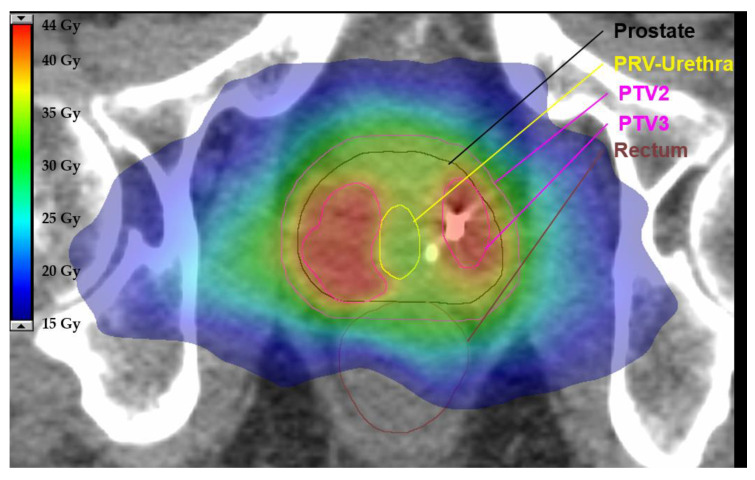
Exemplary axial slide of a representative plan using stereotactic body radiotherapy (SBRT) for the experimental arm. This example demonstrates how overlapping parts between the planning target volume (PTV) and planning organ at risk volume (PRV) are removed to define the final PTV. There is a steep dose gradient to spare the urethra while preserving a sufficient dose to the PTV. Relevant organ as risks, PTVs and isodoses in colorwash are illustrated according to the legend.

**Table 1 cancers-13-05795-t001:** Eligibility Criteria.

**Inclusion Criteria:**
Histologically confirmed adenocarcinoma of the prostate (histological confirmation can be based on tissue taken at any time, but a re-biopsy should be considered if the biopsy is more than 12 months old)
Primary localized PCa (cN0 and cM0 in mpMRI and PSMA-PET): high- or very high-risk according to NCCN v2.2021 OR unfavorable intermediate-risk disease according to NCCN v2.2021
Signed written informed consent for this study
Age > 18 years
Previously conducted PSMA-PET/CT and mpMRI scans or PSMA-PET/MR for staging, fulfilling standard requirements for PCa
ECOG performance score 0 or 1
IPSS score ≤ 15
Prostate volume ≤ 75 mL at RT planning
**Exclusion criteria:**
Evidence of neuroendocrine tumor cells
Prior radiotherapy to the prostate or pelvis
Prior radical prostatectomy
Prior focal therapy approaches to the prostate
Time gap between the beginning of ADT and conduction of initial mpMRI and PSMA-PET scans is >1 month
Radiologically suspicious or pathologically confirmed lymph node involvement (cN+) in mpMRI and/or PSMA-PET/CT
Evidence of metastatic disease (cM+) in mpMRI and/or PSMA-PET/CT
Evidence of cT4 disease in mpMRI and/or PSMA-PET/CT
PSA > 30 ng/mL prior to starting ADT
Expected patient survival <5 years
Bilateral hip prostheses or any other implants/hardware that would introduce substantial CT artefacts
Contraindication to undergo a mpMRI scan
Prostate surgery (TURP or HOLEP) with a significant tissue cavity or prostate surgery (TURP or HOLEP) within the last 6 months prior to randomization
Medical conditions likely to make radiotherapy inadvisable, e.g., acute inflammatory bowel disease, hemiplegia or paraplegia
Previous malignancy within the last 2 years (except basal cell carcinoma or squamous cell carcinoma of the skin), or if previous malignancy is expected to significantly compromise 5 year survival
Any other contraindication to external beam radiotherapy (EBRT) to the pelvis
In mpMRI and PSMA-PET/CT or PSMA-PET/MRI scans, no visible tumor
Participation in any other interventional clinical trial within the last 30 days before the start of this trial
Simultaneous participation in other interventional trials that could interfere with this trial; simultaneous participation in registry and diagnostic trials is allowed
Patient without legal capacity who is unable to understand the nature, significance and consequences of the trial
Known or persistent abuse of medication, drugs or alcohol
Patients expected to have severe set up problems
Dose constraints for organs at risk cannot be adhered to

**Table 2 cancers-13-05795-t002:** Prescription doses of the target volumes (PTVs) for stereotactic body radiotherapy (experimental arm).

PTV1 (PTV1–PTV3)		CTV1	PTV2 (PTV2–PTV3)		CTV2	PTV3		
D50%	D98%	D99%	D50%	D98%	D99%	D50%	D98%	D0.01cc
≥30 Gy	≥28.5 Gy	≥28.5 Gy	≥35 Gy	≥33.25 Gy (minor deviation ≥ 31 Gy)	≥34.5 Gy (minor deviation: ≥34 Gy)	40–42 Gy	≥95% of prescribed dose (minor deviation: up to 36.25 Gy)	≤105% of prescribed dose (minor deviation up to 110%)

**Table 3 cancers-13-05795-t003:** Dose constraints for organs at risk for stereotactic body radiotherapy in 5 fractions (experimental arm). PRV-Rectum_adapt is created by 2 mm isotropic expansion of Rectum_adapt. Rectum_adapt will be generated by delineation of the rectum in the co-registered CB-CTs, MV-CTs or MRI images of the first three fractions. Manual delineation of the Rectum_adapt should consider the mean rectal volume.

OAR	Constraint
PRV-Rectum or PRV-Rectum adapt	D0.03cc (near Dmax): <38 Gy (minor deviation: 38–40 Gy)
Rectum	D1cc: <36 Gy (minor deviation: 36–38 Gy)
	D2cc: <35 Gy
	D20%: ≤28 Gy
PRV-Urethra	D0.01cc (near Dmax): <40 Gy (minor deviation: 40–42 Gy)
Urethra	D50%: <36 Gy (minor deviation: 36–39 Gy)
Bladder	D0.03cc: <38.06 Gy (minor deviation: 38.06–40 Gy)
	D5cc: <37 Gy
	D15%: ≤32 Gy:
	D20%: ≤28 Gy
	D50%: <18.12 Gy (minor deviation: 18.12–20 Gy)
Penile bulb (facultative)	D90%: ≤20 Gy
	D50%: ≤29.5 Gy
Femoral head (left or right)	D5%: ≤28 Gy
Small bowel	D < 5cc: 18.1 Gy
	D0.01cc (near Dmax): <30 Gy (minor deviation: 30–33 Gy)

**Table 4 cancers-13-05795-t004:** Recommendations for the prescription doses of the target volumes (PTVs) for moderate hypofractionation (control arm).

	PTV1		PTV2			CTV2
Prescription dose	D95%	D98%	D95%	D98%	D2%	D98%
60 Gy	95%	≥42 Gy	95%	≥55 Gy	≤61.2 Gy (minor deviation up to 63 Gy)	≥58.8 Gy
62 Gy	95%	≥42 Gy	95%	≥57 Gy	≤63.2 Gy (minor deviation up to 65 Gy)	≥58.8 Gy

**Table 5 cancers-13-05795-t005:** Dose constraints for organs at risk for moderate hypofractionation in 20 fractions (control arm). Rectum_adapt will be generated by delineation of the rectum in the co-registered CB-CTs, MV-CTs or MRI images of the first three fractions. Manual delineation of the Rectum_adapt should consider the mean rectal volume.

Organ	Dose for 20 Fractions (Gy)	Max. Volume (%) Optimal	Max. Volume (%) Mandatory
Rectum or Rectum Adapt	Dose for 20 Fractions (Gy)	Max Volume (%)	Max Volume (%)
	20		85
	25		80
	30		57
	40	38	50
	50		22
	57	5	10
	60	0.01	0.3
Bladder	Dose for 20 fractions (Gy)	Max volume (%)	Max volume (%)
	40	30	40
	50	15	25
	60	3	5
Femoral head left and right	Dose for 20 fractions (Gy)		Max volume (%)
	41		50
Penile bulb	Dose for 20 fractions (Gy)	Max volume (%)	Max volume (%)
	41	50	
	49	20	
Sigmoid	Dose for 20 fractions (Gy)	Max volume (%)	Max volume (cc)
	53	3	
Small bowel	Dose for 20 fractions (Gy)		Max volume (cc)
	41		17
	47		Dmax

## Data Availability

Not applicable.

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
