# Peer review of "PSMA-PET/MRI-Based Focal Dose Escalation in Patients with Primary Prostate Cancer Treated with Stereotactic Body Radiation Therapy (HypoFocal-SBRT): Study Protocol of a Randomized, Multicentric Phase III Trial"

_cancers, 2021, doi:10.3390/cancers13225795_

Round 1

Reviewer 1 Report

The Authors present an interesting study protocol of a phase III trial implementing most modern imaging techniques and RT approaches. They compare the standard RT treatment with an RT concept of reduced treatment time and focal dose escalation based on the patient’s individual tumor geometry. This HypoFocal-SBRT trial aims to personalize treatment of prostate cancer patients .

The Authors will compare the standard of care (a moderated hypofractionation) to two modern RT concepts in the experimental arm of the study: (i) stereotactic body RT (SBRT) and (ii) focal dose-escalated RT based on advanced medical imaging.

Target identification is based on PSMA PET/CT and mpMRI.

I would suggest adding a figure summarizing the study.

I would suggest to better define primary endpoints

Article title: too long I would recommend shortening. Moreover, it seems to focus on IGRT. Imaging is crucial for delineation in the present study, however the concept of IGRT may be more related to imaging for treatment delivery. I think that the title need to be reviewed in order to be more understandable

Graphical abstract: the figure is not complete 1 image is cut.

Minor

Page 3 line 58 In moderately hypofractionated RT (MHRT) RT… put a comma after (MHRT)

Page 3 line 64 : 6.1 – 7-25 Gy per fraction typos 7.25 Gy

Figure 3: rektum, correct to rectum

Reviewer 2 Report

This paper by Zamboglou, et al. report on a new radiation based prostate cancer treatment protocol.  Please see comments below.

  1. For the graphical abstract, I would recommend the authors choose sharper images.
  2. In the study design, the radiation doses in 2.1 are not exact, e.g. ... 40-42 Gy or 60-63 Gy), why is the dose a range rather than an exact value? Could the authors kindly explain.
  3. Figure 3 looks very complicated. Would there be a better way to show this? Maybe divide them in different figures especially for the part where multiple organs overlay.
  4. The caption for the Tables and Figures are not written in a proper formal format. In addition some of sections have the "red" squiggles under them. Please correct. These sections makes the manuscript very informal.

Reviewer 3 Report

This is an excellent protocol combining newly introduced PSMA-PET and focal stereotactic radiation therapy in prostate cancer. As the authors referred in the discussion, this study will reveal dose-effect relationship of prostate cancer and potential implications of PSMA-PET in therapeutic intervention. I hope the study will be launched and wait for the results. 

Round 2

Reviewer 2 Report

Thank you for making the changes. The manuscript is better.